# Placental Pathology and Placental Growth Factor (PlGF)/Vascular Endothelial Growth Factor Receptor-1 (VEGFR-1) Pathway Expression Evaluation in Fetal Congenital Heart Defects

**DOI:** 10.3390/life15060837

**Published:** 2025-05-22

**Authors:** Alexandru Cristian Bolunduț, Ximena Maria Mureșan, Rada Teodora Suflețel, Lavinia Patricia Mocan, Simina Pîrv, Sergiu Șușman, Carmen Mihaela Mihu

**Affiliations:** 11st Department of Pediatrics, “Iuliu Hațieganu” University of Medicine and Pharmacy, 400370 Cluj-Napoca, Romania; 2Department of Histology, “Iuliu Hațieganu” University of Medicine and Pharmacy, 400012 Cluj-Napoca, Romania; sufletel_rada@yahoo.com (R.T.S.); lavinia.trica@gmail.com (L.P.M.); serman_s@yahoo.com (S.Ș.); carmenmihu2004@yahoo.com (C.M.M.); 3Personalized Medicine and Rare Diseases Department, MEDFUTURE—Institute for Biomedical Research, “Iuliu Hațieganu” University of Medicine and Pharmacy, 400349 Cluj-Napoca, Romania; ximena.muresan@medfuture.ro (X.M.M.); simina.pirv@medfuture.ro (S.P.); 4Laboratory of Pathology, IMOGEN Research Center, County Emergency Clinical Hospital, 400012 Cluj-Napoca, Romania

**Keywords:** congenital heart defects, placenta, pathology, placental growth factor, vascular endothelial growth factor receptor-1

## Abstract

The heart and placenta have simultaneous embryologic development, the interactions between the two organs representing the heart–placental axis. They both share key developmental pathways, one of which involves the placental growth factor (PlGF) and its receptor, vascular endothelial growth factor receptor-1 (VEGFR-1). The aim of this study was to evaluate the placental pathology and the expression patterns of PlGF and VEGFR-1 in pregnancies with fetuses with congenital heart defects (CHDs). We analyzed placental gross and microscopic alterations between placentas from pregnancies with CHD fetuses and pregnancies with structurally normal heart fetuses. We also performed the immunohistochemical (IHC) assessment of the placental expression of PlGF and VEGFR-1 in the two groups. We discovered significant gross placental abnormalities in pregnancies with CHD fetuses, including a shorter umbilical cord, marginal or velamentous umbilical cord insertion, and a lower fetal-to-placental weight ratio. Also, 88.2% of the placentas in the CHD group displayed microscopic pathologic aspects. We demonstrated significant placental immunostaining for PlGF and VEGFR-1 in the syncytiotrophoblast and decidual cells compared to villous endothelial cells. We identified a lower placental IHC expression of PlGF in pregnancies with CHD fetuses compared to controls but no differences in the placental immunostaining pattern for VEGFR-1 between the two groups. Our study uncovered a potential role played by the PlGF/VEGFR-1 pathway in the development of CHDs through placental-mediated mechanisms.

## 1. Introduction

The global burden of congenital heart defects (CHDs) is increasing, reaching a prevalence of 9.41 per 1000 births, becoming about 10% higher in the last 15 years, and being one of the most common types of congenital defects [1]. The normal development of the cardiovascular system is orchestrated by interactions between a great number of genes, coding for transcription factors and morphogenetic proteins [2,3]. CHDs represent a heterogeneous group of disorders, arising from errors in this complex process, causing structural alterations of the anatomy of the heart and the great vessels [3]. Although intense research is directed toward investigating the etiopathogenetic mechanisms involved, more than half of the cases of CHDs cannot be explained by known genetic factors or by environmental exposures and other non-genetic contributors [4].

The placenta is a highly structured vascular organ, linking the fetus to the mother. It plays a critical role in influencing organogenesis through a number of functions, including providing oxygen and nutrients from the mothers’ circulation to promote growth, clearing the fetal circulation of metabolic byproducts, protecting the fetus from hostile aggressions (such as toxins, infections, or oxidative stress), and regulating developmental processes through metabolic, hormonal, and paracrine mediators [5,6].

The heart and placenta have a simultaneous embryologic evolution, with the formation of the beating heart tube being completed at around 21 days after conception, at the same time that the rudimentary villous tree is forming in the placenta [7,8]. The parallel development and interactions between the two organs represent the heart–placental axis, both sharing key developmental pathways, with deleterious effects in any of these processes resulting in morphological and functional perturbations in both the placenta and the cardiovascular system [9,10,11]. Observational studies reported that placental pathology is more common in pregnancies carrying fetuses with CHDs (including low placental weight, reduced placental efficiency, fibrin deposits, thrombosis, signs of fetal and maternal vascular malperfusion), and placental abnormalities (such as the abnormal insertion of the umbilical cord, preeclampsia) may increase the risk for CHDs [6,8,12]. Based on single-cell sequencing techniques, it was found that more than 300 genes are commonly expressed between first-trimester heart and placental endothelial cells, and 16 to 53 genes are commonly expressed between first-trimester cardiomyocytes and other placental cell types (extravillous trophoblast, cytotrophoblast, syncytiotrophoblast) [13]. The canonical Wnt/β-catenin pathway, folate metabolism, NOTCH signaling pathway, and GATA family of transcription factors, especially *GATA4*, are just a few examples of the common determinants of the normal development of both the placenta and the heart [7,8,10,11,14].

One of the common developmental pathways between the cardiovascular system and the placenta involves the vascular endothelial growth factor (VEGF) family, which are key regulators of angiogenesis [14]. The placental growth factor (PlGF) is a member of this family of growth factors that was first identified in human placental tissues but is also present in the uterine mucosa, heart, lungs, skeletal muscle, adipose tissue, and skin cells [15,16]. The PlGF determines its effects through binding with high affinity to the VEGF receptor-1 (VEGFR-1), formerly known as Flt-1, a transmembrane tyrosine-kinase receptor [17]. In the placenta, the growth factor seems to play a role in trophoblast growth and differentiation, trophoblast invasion, and the formation of the vascular network of the developing villous tree [15,16,18]. In addition to that, a recent study identified a dual role for the PlGF in heart development, demonstrating both cardiomyogenic and vasculogenic effects [19]. These pleomorphic actions make the growth factor a plausible mediator in the heart–placenta axis.

The aim of this study was to evaluate the placental pathology in pregnancies carrying fetuses with congenital heart defects and to determine the expression patterns of PlGF and VEGFR-1 in placentas from these pregnancies compared to placentas from pregnancies without this type of congenital defect. We hypothesized that the abnormal development of the heart may be associated with altered placental expression patterns for the growth factor and its receptor, thus investigating a potential pathogenic mechanism.

## 2. Materials and Methods

### 2.1. Study Population

We conducted a single center retrospective pilot study. Clinical and pathological data were obtained from the Laboratory of Pathology, IMOGEN Research Center, County Emergency Clinical Hospital (Cluj-Napoca, Romania). Cases were randomly selected from samples referred to the Laboratory of Pathology between 2018 and 2022, from pregnancies terminated in miscarriage, therapeutic abortion, or stillbirth, with a confirmed congenital heart defect at the pathology examination of the fetus. For each case, a gestational age-matched control was selected from the samples from the Laboratory of Pathology in the same period of time, from pregnancies terminated in miscarriage, therapeutic abortion, or stillbirth, with a confirmed structurally normal heart at the pathology examination of the fetus. Multiple gestation pregnancies, fetuses with genetic abnormalities, and pregnancies complicated with maternal diabetes, preeclampsia, maternal hypertension, or maternal coagulopathy were excluded from both groups.

This retrospective study was approved by the Ethics Committees of “Iuliu Hațieganu” University of Medicine and Pharmacy, Cluj-Napoca, Romania (AVZ270/2023) and County Emergency Clinical Hospital, Cluj-Napoca, Romania (40721/2023). This study followed the ethical guidelines of the Declaration of Helsinki. We mention that prior to the pathological analysis, during the hospitalization, patients signed an informed consent form for the storage and use of the samples in future studies.

### 2.2. Maternal and Fetal Data Collection

Maternal data, including the age and number of previous gestations, were collected by reviewing the respective electronic files from the Obstetrics and Gynecology Clinics, County Emergency Clinical Hospital (Cluj-Napoca, Romania).

Fetal data were collected retrospectively based on routine clinical pathology reports issued at the time of pregnancy termination from the electronic database of the Laboratory of Pathology, IMOGEN Research Center, County Emergency Clinical Hospital (Cluj-Napoca, Romania). Gestational age and anthropometric data, including body weight, crown–heel length, crown–rump length, and head circumference were recorded. Z-scores for these parameters were calculated based on published fetal and neonatal autopsies quantitative standards [20]. Fetal pathology reports were reviewed for confirmation of the presence of CHDs or a structurally normal heart as the base for the inclusion of subjects in one of the two groups, cases and controls, and the presence of other congenital anomalies was also recorded.

### 2.3. Placental Gross and Microscopic Evaluation

For all the subjects included in this study, gross placental evaluation after formalin fixation was performed at the time of pregnancy termination, based on routine clinical practices and published guidelines [21]. The issued pathology reports were reviewed. The evaluation included measurements of placental weight, length, width, and thickness, the aspect, insertion, and dimensions of the umbilical cord, and other gross pathological findings, such as, infarction, thrombi, or calcifications. The insertion of the umbilical cord was considered marginal if it was less than 2 cm from the nearest placental margin, velamentous if inserted directly on the membranes, and eccentric if inserted more than 2 cm from the nearest margin, but not central. A hypercoiled umbilical cord was defined as more than 3 coils per 10 cm [21].

The microscopic examination of placental tissue samples after hematoxylin and eosin staining was also performed at the time of pregnancy termination, based on the published guidelines [21]. The descriptions from the pathology reports were analyzed and information regarding the presence of chorangiosis, decidual arteriopathy, thrombosis, subchorionic hemorrhage, fibrin deposits, chorioamnionitis, infarction, or calcifications was recorded.

### 2.4. Placental Immunohistochemical Evaluation

Paraffin-embedded placental tissue samples were available for some of the subjects included in this study from the archives of the Laboratory of Pathology, IMOGEN Research Center, County Emergency Clinical Hospital (Cluj-Napoca, Romania). Standard immunohistochemical (IHC) staining was performed. Briefly, 4 µm sections were dewaxed and rehydrated through serial incubations in xylene, 100-95-90% ethanol and water, followed by antigen retrieval in citrate buffer solution (Bio SB, Santa Barbara, CA, USA). Nonspecific binding sites were blocked by incubation in 5% bovine serum albumin (BSA) solution (Sigma-Aldrich, Saint Louis, MO, USA) for 1 h at room temperature (RT). Tissues were then incubated for 1 h at RT with primary antibodies against PlGF (Proteintech, Rosemont, IL, USA, 10642-1-AP, 1:100) and VEGFR-1 (Proteintech, Rosemont, IL, USA, 13687-1-AP, 1:500), followed by 30 min of RT incubation with ready-to-use horseradish peroxidase (HRP)-conjugated secondary antibody (Proteintech, Rosemont, IL, USA, RGAR011), and finally, a 10 min incubation with 3,3-diaminobenzidine (DAB) substrate (Dako Agilent, Carpinteria, CA, USA). IHC negative controls were also performed by staining with secondary antibody only and omitting the step of primary antibody incubation. Tissues were then counterstained with Mayer’s Hemalum solution (Merck Millipore, Darmstadt, Germany), followed by gradient dehydration in 95–100% ethanol and mounting with Sub-X mounting medium (Electron Microscopy Sciences, Hatfield, PA, USA). Sections were randomized and blindly analyzed separately by two specialists for the distribution and the overall intensity of the immunostaining, using a four-level semiquantitative scale: 0 for no staining, 1 for weak, 2 for moderate, and 3 for strong staining. Differences were resolved through a detailed examination of the particular fields by both experts, leading to a mutually accepted classification. Images were acquired using the Aperio LV1 IVD microscope slide scanner (Leica Biosystems, Richmond, IL, USA).

### 2.5. Statistical Analysis

Quantitative data were first analyzed to establish normality using the Shapiro–Wilk test. Data with non-normal distribution were presented as median [quartile 1–quartile 3], and appropriate nonparametric (Mann–Whitney and paired Wilcoxon) tests were performed. Correlations between quantitative variables were assessed using the Spearman test. Qualitative data, including immunostaining distribution, were expressed as proportions, and associations were tested using Fisher’s exact test and the Cochran–Armitage test for trend, respectively. All statistical analyses were performed using R Software (version 3.1.1; The R Foundation, Vienna, Austria) and GraphPad Prism Software (version 8.0.1; GraphPad Software Inc., La Jolla, CS, USA). Differences were considered significant at a *p* < 0.05.

## 3. Results

### 3.1. Characteristics of the Study Groups

A total of 102 subjects were included in this study, 51 CHD cases and 51 gestational age-matched controls, with a median gestational age of the pregnancies of 20 weeks (Q1–Q3: 16–23 weeks). There were no statistically significant differences between the two groups in terms of maternal age—median age of 31 years [27–36 years] in the CHD group vs. 30 years [28–34 years] in controls; *p* = 0.4102—and number of previous pregnancies—32 primigravida (62.7%) in the CDH group vs. 30 (58.8%) in controls; *p* = 0.8394.

Fetal pathology and anthropometric data are presented in Table 1 and Figure 1. No statistically significant difference was observed between sex distribution in the two groups. The prevalence of other congenital defects was higher in pregnancies complicated with heart defects. In both groups, central nervous system defects, including holoprosencephaly, the agenesis of the corpus callosum, polymicrogyria, ventriculomegaly, Arnold–Chiari malformation, and neural tube defects, were the most commonly associated congenital anomalies, with no significant difference in frequency between them—15 subjects (29.4%) in the CDH group vs. 14 subjects (27.5%) in controls; *p* > 0.9999. The atrioventricular septal defect was the most common congenital heart defect in the case group (11 cases—21.6%), followed by the ventricular septal defect and hypoplastic left heart syndrome (both with 8 cases each—15.7%), and the double-outlet right ventricle (6 cases—11.8%) (Figure 1). Fetal anthropometric parameters, including weight, crown–heel length (CHL), and crown–rump length (CRL), were significantly decreased in the CHD pregnancies compared to controls, both in absolute measurement and Z-score, with the exception of absolute weight. The head circumference (HC) did not differ between the two study groups (Table 1).

### 3.2. Morphological Characteristics of the Placenta and the Umbilical Cord

The pathological evaluation of the umbilical cord (UC) is described in Table 2. The median length of the UC was significantly shorter in pregnancies carrying fetuses with CHD compared to controls. The UC length was moderately correlated with both fetal weight—Spearman r = 0.6932 (95% CI = 0.5718–0.7849); *p* < 0.0001—and placental weight—Spearman r = 0.5894 (95% CI = 0.4397–0.7072); *p* < 0.0001. The marginal (less than 2 cm from the nearest border) or velamentous insertions of the UC were associated with an increased risk for CHD development—odds ratio (OR) = 4.38 (95% CI = 1.78–11.23)—while central insertion of the UC was significantly associated with a lower risk for CHDs—OR = 0.14 (95% CI = 0.04–0.41). Also, the presence of a single umbilical artery was associated with an increased risk for CHDs—OR = 4.85 (95% CI = 1.19–28.69). No differences between the two groups were observed in terms of the UC diameter or its hypercoiled aspect (Table 2).

Table 3 presents the gross and microscopic description of the placentas of the investigated pregnancies. In our cohort, fetal weight was strongly correlated with placental weight—Spearman r = 0.8935 (95% CI = 0.8437–0.9280); *p* < 0.0001. Although there were no statistically significant differences between the two groups in terms of placental weight and dimensions and in terms of fetal weight (Table 1), the fetal-to-placental weight ratio was significantly higher, while the placental-to-fetal weight ratio was significantly lower in the control group compared to pregnancies carrying fetuses with CHDs. The microscopic evaluation of the placentas described pathologic aspects in 88.2% of the CHD group, including villous vascular congestion, maternal vascular malperfusion lesions, such as infarction, subchorionic hemorrhage, and decidual arteriopathy, fetal vascular malperfusion lesions, such as fibrin deposits, thrombosis, and chorangiosis, calcifications, or pathologic findings suggestive for chorioamnionitis. With the exception of calcifications, which were more frequent in the CHD group, and villous thrombi, with a higher frequency in the control group, none of the other findings were significantly different between the two groups in our study population.

### 3.3. Immunohistochemical Study

For 20 CHD cases and their respective gestational age-matched controls, paraffin-embedded placental tissue samples were available for the immunohistochemical evaluation of the distribution and expression of the PlGF and VEGFR-1. The staining showed a similar cytoplasmic immunolocalization of the investigated angiogenic factor and its receptor, both being detected in the syncytiotrophoblast, decidual cells, and endothelial cells of the central villous vascular axis (Figure 2, Appendix A).

Regarding IHC staining intensity assessment, a correspondence of 84.2% was met between the two independent examiners. The PlGF displayed a significantly weaker overall intensity of the immunostaining in all the investigated structures compared to the VEGFR-1 (Table 4). Stronger immunoreactivity for the PlGF was demonstrated at the level of both syncytiotrophoblast and decidual cells compared to villous endothelial cells, the first displaying the strongest staining of all, although not statistically significant compared to decidual cells (Figure 3a and Appendix A). A similar pattern of staining intensity was observed regarding the VEGFR-1, with significantly stronger staining localized in the syncytiotrophoblast and decidual cells compared to villous endothelial cells, but the highest expression was observed in decidual cells, although not statistically significant compared to the syncytiotrophoblast (Figure 3b and Appendix A).

When comparing the placental immunostaining intensities between the CHD group and the control group (Table 5), there was a significantly lower staining intensity of the PlGF in the syncytiotrophoblast and decidual cells in the placentas from pregnancies with CHD fetuses. No difference was observed at the level of the villous endothelial cells regarding PlGF staining between the two groups. When stratified by CHD type (Appendix A), the same significantly lower PlGF staining pattern in the syncytiotrophoblast was observed in placentas from pregnancies with atrioventricular septal defect (AVSD) and ventricular septal defect (VSD) fetuses, but not in hypoplastic left heart syndrome (HLHS) cases. Also, AVSD cases displayed a significantly lower PlGF staining at the level of decidual cells compared to controls, but this difference was not observed in the other investigated CHD types. The VEGFR-1 displayed no statistically significant differences in the staining pattern between the CHD and the control group, with similar findings when stratified by CHD type (Table 5 and Appendix A).

## 4. Discussion

The cardiovascular system and the placenta are interlinked not only in terms of developmental chronology but also in terms of signaling pathways and regulatory mechanisms, each organ, in part, mediating the proper development of the other. This represents the heart–placenta axis. Our study aimed to explore this interaction by evaluating the placental pathology and expression patterns of a common signaling pathway, the one mediated by the PlGF and VEGFR-1, in pregnancies carrying fetuses with congenital heart defects.

One of the placenta’s main roles is to provide oxygen and nutrients from the mother’s circulation for fetal growth [5]. Our study showed significantly altered fetal growth in the CHD group, with a shorter crown–heel length and crown–rump length (both absolute values and Z-scores) and lower weight Z-scores (the absolute value was still lower, but it did not reach statistical significance). These results confirm those reported by other researchers. O’Hare et al. revealed significantly lower birthweight and birth length in CHD fetuses [21]. Matthiesen et al. validated the lower birthweight Z-scores of CHD cases compared to healthy controls in a large nationwide Danish cohort [22], while Jones et al. confirmed the findings in a hypoplastic left heart syndrome (HLHS) cohort [23]. Since the median gestational age of the subjects included in our study was lower than in the abovementioned studies, we can conclude that the interactions between an abnormally developed heart and the placenta contribute to altered somatic growth by the earlier stages of fetal development.

Pregnancies carrying fetuses with CHDs tend to have lower placental weight (PW) relative to their gestational age on both pathology and magnetic resonance imaging (MRI)-based evaluation, particularly in the transposition of great arteries (TGA) and HLHS cases [23,24,25,26], although not all the studies reached statistical significance [21,27]. The mentioned MRI-based study demonstrated significantly lower total brain and cerebral volumes and higher brainstem volumes in CHD pregnancies [27]. Multiple studies also observed a significantly decreased head circumference in fetuses with CHDs [21,22,24], with TGA cases being associated with a lower head circumference relative to their birthweight (BW) [22]. This observation was not confirmed in our cohort, where the fetal head circumference (both in absolute value and Z-score) did not differ significantly between the two groups. Although no correlation was made between placenta and brain volumes [27], it was suggested that decreased placental growth was associated with impaired fetal growth, which may also impact neurodevelopment, particularly in CHD fetuses.

Placental efficiency is defined as grams of fetus produced by a gram of placenta [28], and most of the studies estimate it by using either the fetal-to-placental weight ratio or the placental-to-fetal weight ratio [21,22,25,29,30,31]. Conflicting results were reported regarding the placental efficiency in relation to CHDs. While most of the researchers revealed no significant differences in the fetal-to-placental weight ratio in CHD cases [31,32], two retrospective studies suggest higher placental efficiency in CHD pregnancies [25,29]. A reduced placental-to-fetal weight ratio could signify more efficient nutrient transfer from placental circulation to the fetus, meeting the increased demands from the fetal pathologic condition [6,8,26]. Although no significant differences were observed in our study regarding the absolute value of fetal weight and placental weight in the CHD group compared to controls, the fetal-to-placental weight ratio displayed differences that were statistically significant between the two investigated groups (the fetal-to-placental weight ratio was lower in the CHD group), in contradiction with the abovementioned literature findings. Most of the previously mentioned studies evaluated term pregnancies, so there is little evidence on the dynamics of these ratios at other specific moments during pregnancy. Because of the complex interaction between the fetus and the placenta [33], further studies are needed in order to accurately evaluate the relationship in fetus growth among particular placental conditions.

The umbilical cord (UC) is the link between the placenta and the fetus, and its abnormal development can produce changes in blood flow and disturbances in nutrient delivery, which may affect the microenvironment of the fetus [31]. The length of the UC was positively correlated with birthweight in previous studies [30], an observation that was also confirmed by our analysis. Based on our data, the median length of the UC was significantly shorter in pregnancies carrying fetuses with CHDs, partially accounting for the lower fetal weight. Abnormal cord insertion was also associated with adverse pregnancy outcomes, such as preterm birth, small-for-gestational-age infants, low-birthweight infants, and intrauterine fetal death [6,26]. Albalawi et al. revealed that abnormal cord insertion (marginal or velamentous) is associated with an increased risk of developing CHDs (OR = 2.33–3.76), with the highest association with conotruncal defects [31]. Other studies also describe higher rates, ranging between 16.4 and 19%, depending on the size of the investigated cohort [24,34]. Similar findings were presented in our study, where the marginal or velamentous insertion of the UC was significantly associated with CHDs, increasing the risk of abnormal heart development by 4.38 times. The presence of a single umbilical artery was significantly associated with a 4.85 times increase in risk for CHDs; also, in accordance with previously published data, Leon et al. reported this UC abnormality in 55% of CHD pregnancies [24]. The hypercoiled aspect of the UC was described in relation with CHDs, in a proportion of 28–32%, but studies lack a comparison with a control group [24,34]. Our data found a higher frequency of hypercoiled UC (45%) but revealed no significant association with CHDs.

The Amsterdam Placental Workshop Group Consensus statement unified the sampling and definitions of placental lesions. Based on that, maternal vascular malperfusion (MVM) is defined as injury caused by alterations in the uterine and intervillous flow from the maternal circulation, with pathology indicators including infarcts, retroplacental hemorrhage, distal villous hypoplasia, accelerated villous maturation, and decidual arteriopathy, while fetal vascular malperfusion (FVM) consists of lesions resulting from abnormal fetal perfusion to the villous parenchyma and are represented by thrombosis, avascular villi, fibrin deposits, and chorangiosis [6,35]. Previous studies reported a high prevalence of these histopathologic lesions in placentas from pregnancies with CHD fetuses, with numbers that vary between 78% and 85.7% [24,36]. These results are similar to the findings from our study, where 88.2% of the placentas in the CHD group carried at least one pathological aspect. O’Hare et al. demonstrated significantly higher rates of MVM and delayed villous maturation in term pregnancies with fetal CHDs, with the highest association in two-ventricle anatomy heart lesions, but not FVM, suggesting that placental maldevelopment may be associated with maternal factors [21]. On the other hand, Rakha et al. reported predominantly FVM lesions in term placentas from pregnancies with CHD fetuses, particularly chorangiosis, present in one-third of the subjects included in their study [36]. Chorangiosis represents an increase in capillary density in villous tissue and was suggested to reflect prolonged low-grade placental hypoxia [36]. The findings were also confirmed by Rychik et al., particularly reporting thrombosis and chorangiosis, with the highest proportion in the transposition of the great arteries subjects [25], while Jones et al. revealed increased fibrin deposits in hypoplastic left heart syndrome subjects [23]. Our data showed relatively similar proportions of both MVM and FVM lesions in the CHD group (52.94% and 68.62%, respectively). In addition, the CHD group had a significantly higher rate of placental calcifications compared to controls. All these contradicting findings underline the complex nature of the pathological mechanisms involved in the relation between the CHD and placenta.

The VEGF family and their receptors seem to play an important role in the processes of placental vasculogenesis (the formation of endothelial progenitor cells from the extraembryonic mesoderm) and angiogenesis (the formation of new blood vessels from existing ones) [16,17,18]. The placental growth factor (PlGF) is a key member of the abovementioned family of growth factors in the context of placental development, being highly expressed in the placenta during all stages of gestation [15]. PlGF effects are mediated through its interaction with the tyrosine-kinase receptor VEGFR-1 [15,16]. Also, it may enhance the effects of other molecules from the same family, especially VEGF-A, by binding to VEGFR-1 and allowing for more interactions of the other molecules with VEGFR-2, thus potentiating the effects mediated by this other receptor [15,16,37].

Previous studies localized PlGF expression within the villous trophoblast, especially the syncytiotrophoblast, the endothelial cells of villous capillaries, and the media of the large vessels of the placenta [15,38,39,40,41], while VEGFR-1 was distributed mainly in the syncytiotrophoblast and endothelial cells [23,38,39]. Our results showed the same immunolocalization, while also displaying positive IHC staining in decidual cells for both epitopes. Moreover, based on our semiquantitative IHC analysis, PlGF expression was the highest in the syncytiotrophoblast. Regarding VEGFR-1 expression, both syncytiotrophoblast and decidual cells displayed significantly stronger staining compared to endothelial cells, with no statistically significant difference between them, although the proportion of moderate- and strong-stained areas was higher for decidual cells. Our data confirmed the observation of Ehrlich et al., which was that the overall staining for PlGF is weaker than the staining for VEGFR-1 [38], demonstrating a statistically significant difference in the staining intensity in all of the three areas investigated: syncytiotrophoblast, decidual cells, and villous endothelial cells.

This localization of PlGF expression suggests its involvement in trophoblast growth and differentiation, trophoblast invasion and blastocyst implantation, and nonbranching angiogenesis [16,18,37]. Experimental in vitro studies conducted by Wu et al. on trophoblast cell lines demonstrated a significant decrease in *PGF* gene expression, with the downregulation of genes involved in cell cycle progression (such as *CCNA2*, *CCNB1*, *CCND1*, and *CCNE1*) after the administration of ZM-306416, a selective VEGFR-1 inhibitor. Also, they observed a decrease in the number of cells and their motility, thus underlining the role played by the PlGF/VEGFR-1 pathway in trophoblastic cell proliferation and migration, acting as an autocrine mediator of the trophoblast function [41].

Our study revealed a statistically significant decrease in PlGF expression in the syncytiotrophoblast and decidual cells of placentas from pregnancies with fetuses with CHDs compared to controls, based on IHC staining, but no difference in VEGFR-1 immunostaining distribution. In the subgroup analysis, this difference was present in AVSD and VSD cases. Similar expression patterns were observed in relation to other adverse pregnancy outcomes as well. Kapustin et al. revealed a significant decrease in the placental IHC expression of PlGF, compared to controls, in pregnancies complicated with preeclampsia but also in diabetic pregnancies, the lowest immunostaining being reported in mothers with type 1 diabetes mellitus without preconception planning [40]. Alahakoon et al. showed lower placental PlGF and VEGFR-1 IHC-stained areas and intensities in pregnancies complicated with intrauterine growth restriction (IUGR) with or without preeclampsia compared to controls [39]. Fetal growth restriction (FGR) was also addressed by Wu et al., confirming the lower placental expression of the PlGF in FGR pregnancies by IHC assay but also through ELISA and RT-qPCR, offering a more accurate quantification [41]. Also, the group explained that the downregulation of PlGF expression in FGR placentas was due to epigenetic mechanisms, discovering higher methylation patterns in a CpG island located downstream of exon 7 of the *PGF* gene compared to control placentas [41]. The placental vasculature and hemodynamics play an important role in all these pregnancy adverse outcomes, underlining the importance of the PlGF/VEGFR-1 pathway in regulating its proper development.

Jones et al. evaluated the placental expression of angiogenic factors in pregnancies carrying fetuses with HLHS and demonstrated a significant reduction in the mRNA expression of the PlGF in the presence of the congenital heart defect, but no change in the expression of VEGF-A compared to healthy controls [23]. Although there was no difference in the mRNA expression of the VEGFR-1 between HLHS cases and controls, the immunohistochemical analysis localized the protein’s expression to the fetal endothelium, which appeared reduced in HLHS cases [23]. In our study, the subgroup analysis showed no significant difference in the immunostaining distribution of both the PlGF and VEGFR-1 between HLHS cases and controls.

Circulating PlGF (cPlGF) can be used as a marker of placental dysfunction. Agrawal et al. demonstrated that maternal cPlGF profiles during pregnancy are a better predictor of stillbirth compared to uterine artery Doppler evaluation, and that the combination of the two examinations is associated with specific placental pathology diagnoses [42]. For example, decreasing levels of cPlGF with abnormal uterine artery Doppler waveforms are associated with maternal vascular malperfusion lesions, while persistent low cPlGF values without alterations in uterine artery Doppler waveforms are an indicator of massive perivillous fibrin deposition or chronic histiocytic intervillositis [42]. Fantasia et al. observed lower maternal serum PlGF and pregnancy-associated plasma protein-A (PAPP-A) levels in first-trimester pregnancies with fetuses suffering from CHDs, with no difference in the uterine artery pulsatility index compared to controls, suggesting that even in early pregnancy, there is evidence of placental dysfunction in the absence of altered placental perfusion [43]. Moreover, Sugimoto et al. revealed postnatal changes in the soluble VEGFR-1 (sVEGFR-1) to serum PlGF ratio in CHD children, correlating the ratio with volume overload and with persistent hypoxia, suggesting their involvement in the developed heart as well [44].

Our study is one of the few evaluating the placental expression of the PlGF and its receptor, VEGFR-1, in the context of congenital heart defects. One of the major limitations of our study is the relatively small sample size included. Moreover, the CHD group was heterogeneous, limiting our ability to more accurately examine the relationship between specific types of CHDs and placental pathology. Also, due to the retrospective nature of our study, placental pathology was based on written reports from the moment of pregnancy termination. Although all the samples were examined and the reports were signed by the same specialist, the reporting style varied in time. The overall prevalence of placental pathology may be overestimated by our study because all the subjects were included from pregnancies terminated in miscarriage, therapeutic abortion, or stillbirth, the routine examination of placentas is not a standard of practice, and the decision of sending them to pathology was made by the obstetrician, based on unclear criteria from the available data. Also, the use of immunohistochemical assay for determining protein expression imposes some limitations. In general, IHC is less reproductible compared to other expression evaluation assays, such as Western blot or mRNA expression analysis, in part due to the variability induced by the multi-step process and the experience of the one performing and interpreting the assay. Although the use of polyclonal antibodies, as the ones used in our study, offers a higher sensitivity, it may also lead to a higher nonspecific background due to natural antibodies. IHC offers a semiquantitative evaluation of the antigen’s expression; usually, an arbitrary 4-level intensity scoring system is widely used in most of the studies. The accuracy of this method is low and subject to interpretational variation between examiners. This can be improved by the use of digital image analysis techniques [39] that are starting to emerge both in research and in clinical practice. Future prospective studies are needed in order to address these limitations and validate our preliminary findings regarding the expression patterns of the PlGF and VEGFR-1 by quantitative methods.

In conclusion, our study showed an increased burden of placental abnormalities in pregnancies with congenital heart defect fetuses. We demonstrated a significant placental IHC expression of the PlGF and its receptor, VEGFR-1, in the syncytiotrophoblast and decidual cells. Also, our study identified a lower placental IHC expression of the PlGF in pregnancies with CHD fetuses compared to controls, but no difference in the placental immunostaining pattern for the VEGFR-1 between the two groups. Further studies are needed in order to uncover the mechanisms behind the interrelation between the placenta and the developing heart, to better understand how alterations in vasculogenesis and angiogenesis impact cardiogenesis. This could prove useful not only for understanding the pathogenesis of congenital heart defects but also for creating the possibility of identifying potential diagnostic and prognostic markers, some of them with clinical utility, for example, potentially addressing growth abnormalities, or improving the prognosis of the patients.

## Figures and Tables

**Figure 1 life-15-00837-f001:**
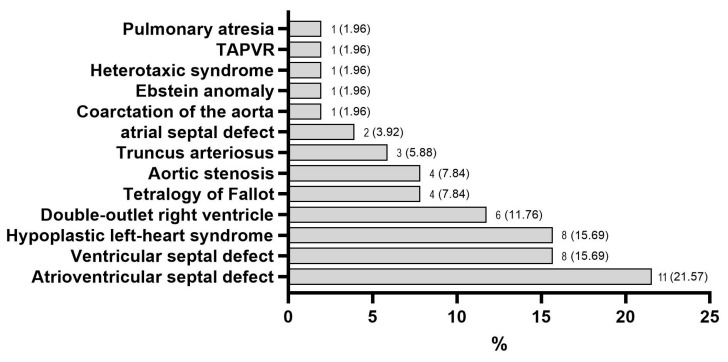
Distribution of the types of congenital heart defects in the studied cohort (TAPVR, total anomalous pulmonary venous return).

**Figure 2 life-15-00837-f002:**
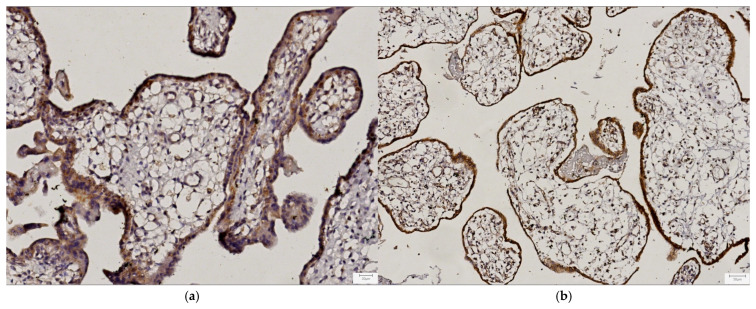
Representative micrographs displaying the placental immunohistochemical localization of the placental growth factor (PlGF) in (**a**) the syncytiotrophoblast (strong staining; magnification 20×), (**c**) decidual cells (strong staining; magnification 20×), and (**e**) villous endothelial cells (moderate staining; magnification 20×) and the vascular endothelial growth factor receptor-1 (VEGFR-1) in (**b**) the syncytiotrophoblast (strong staining; magnification 10×), (**d**) decidual cells (strong staining; magnification 20×), and (**f**) villous endothelial cells (strong staining; magnification 40×).

**Figure 3 life-15-00837-f003:**
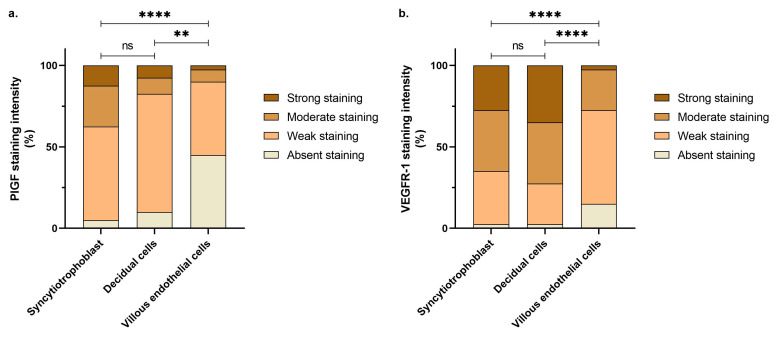
Comparative evaluation of placental immunostaining intensities at the level of syncytiotrophoblast, decidual cells, and villous endothelial cells for (**a**) the placental growth factor (PlGF) and (**b**) the vascular endothelial growth factor receptor-1 (VEGFR-1) (** *p* < 0.001; **** *p* < 0.00001; ns, no significance).

**Table 1 life-15-00837-t001:** Fetal pathology and anthropometric data description in the study population.

Characteristic	CHD (51 Subjects)	Control (51 Subjects)	*p*-Value
Female sex, *n* (%)	23 (45.1)	33 (64.7)	0.0727
Other congenital defects, *n* (%)	38 (74.5)	25 (49.0)	0.0139
Fetal weight (g), median [Q1–Q3]	303 [75.8–599]	321 [123–694]	0.0885 ^‡^
Fetal weight Z-score, median [Q1–Q3]	−0.21 [−1.75–1.57]	0.95 [−0.07–1.9]	0.0283
Fetal CRL (cm), median [Q1–Q3]	16 [10.75–21]	17.5 [12–21.5]	0.0006 ^‡^
Fetal CRL Z-score, median [Q1–Q3]	−0.72 [−2.06–0.4]	0.18 [−0.55–1.06]	0.0017
Fetal CHL (cm), median [Q1–Q3]	23 [15.75–31.63]	25 [18–32.5]	0.0027 ^‡^
Fetal CHL Z-score, median [Q1–Q3]	−0.195 [−1.79–1.2]	0.81 [−0.27–1.94]	0.0098
Fetal HC (cm), median [Q1–Q3]	17.75 [11.88–22.63]	19 [12.5–22.5]	0.1625 ^‡^
Fetal HC Z-score, median [Q1–Q3]	0.21 [−1.11–1.47]	0.42 [−0.08–1.57]	0.2092

CHD, congenital heart defect; CHL, crown–heel length; CRL, crown–rump length; HC, head circumference. ^‡^ paired Wilcoxon test was used (pairing was performed based on gestational age).

**Table 2 life-15-00837-t002:** Umbilical cord pathology description in the study population.

Characteristic	CHD (51 Subjects)	Control (51 Subjects)	*p*-Value
Central UC insertion, *n* (%)	6 (11.8)	25 (49.0)	<0.0001
Eccentric UC insertion, *n* (%)	9 (17.7)	9 (17.7)	>0.9999
Marginal or velamentous UC insertion, *n* (%)	32 (62.8)	14 (27.5)	0.0006
UC length (cm), median [Q1–Q3]	17.25 [11.88–25.88]	25 [18–33.5]	<0.0001 ^‡^
UC diameter (cm), median [Q1–Q3]	0.8 [0.5–1]	1 [0.6–1.2]	0.1616 ^‡^
Hypercoiled * UC, *n* (%)	23 (45.1)	23 (45.1)	>0.9999
Single umbilical artery, *n* (%)	12 (23.5)	3 (5.9)	0.0228

CHD, congenital heart defect; UC, umbilical cord. * more than 3 coils/10 cm. ^‡^ paired Wilcoxon test was used (pairing was performed based on gestational age).

**Table 3 life-15-00837-t003:** Placenta gross and microscopic pathology description in the study population.

Characteristic	CHD (51 Subjects)	Control (51 Subjects)	*p*-Value
Placental weight (g), median [Q1–Q3]	130 [84–187]	137 [90–190]	0.6878 ^‡^
Placental length (cm), median [Q1–Q3]	11 [9.5–13]	12 [10–14]	0.2212 ^‡^
Placental width (cm), median [Q1–Q3]	9.5 [7.25–12]	10 [8–12.5]	0.6874 ^‡^
Placental thickness (cm), median [Q1–Q3]	3 [2–3]	3 [2.5–3]	0.7455 ^‡^
Fetal-to-placental weight ratio, median [Q1–Q3]	2.23 [1.38–3.51]	2.49 [1.66–3.83]	0.0223 ^‡^
Placental-to-fetal weight ratio, median [Q1–Q3]	0.45 [0.28–0.73]	0.4 [0.26–0.6]	0.0141 ^‡^
Villous vascular congestion, *n* (%)	22 (43.1)	26 (51.0)	0.5520
Maternal vascular malperfusion, *n* (%)	27 (52.9)	29 (56.9)	0.8424
Infarction, *n* (%)	8 (15.7)	9 (17.7)	>0.9999
Subchorionic hemorrhage, *n* (%)	24 (47.1)	23 (45.1)	0.8429
Decidual arteriopathy, *n* (%)	6 (11.8)	12 (23.5)	0.1932
Fetal vascular malperfusion, *n* (%)	35 (68.6)	39 (76.5)	0.5061
Thrombosis, *n* (%)	10 (19.6)	23 (45.1)	0.0105
Fibrin deposits, *n* (%)	24 (47.1)	31 (60.8)	0.2332
Chorangiosis, *n* (%)	3 (5.9)	1 (1.96)	0.6175
Calcifications, *n* (%)	11 (21.6)	3 (5.9)	0.0410
Chorioamnionitis, *n* (%)	10 (19.6)	16 (31.4)	0.2558

CHD, congenital heart defect. ^‡^ paired Wilcoxon test was used (pairing was performed based on gestational age).

**Table 4 life-15-00837-t004:** Comparative evaluation of placental immunostaining intensities of the placental growth factor (PlGF) and the vascular endothelial growth factor receptor-1 (VEGFR-1).

Localization	Staining Intensity	VEGFR-1	PlGF	*p*-Value *
Syncytiotrophoblast	Absent staining, *n*	1/40	2/40	0.0164
	Weak staining, *n*	13/40	23/40	
	Moderate staining, *n*	15/40	10/40	
	Strong staining, *n*	11/40	5/40	
Decidual cells	Absent staining, *n*	1/40	4/40	<0.0001
	Weak staining, *n*	10/40	29/40	
	Moderate staining, *n*	15/40	4/40	
	Strong staining, *n*	14/40	3/40	
Villous endothelial cells	Absent staining, *n*	6/40	18/40	0.0046
	Weak staining, *n*	23/40	18/40	
	Moderate staining, *n*	10/40	3/40	
	Strong staining, *n*	1/40	1/40	

PlGF, placental growth factor; VEGFR-1, vascular endothelial growth factor receptor-1. * Cochran–Armitage test for trends.

**Table 5 life-15-00837-t005:** Comparative evaluation of placental immunostaining intensities of the placental growth factor (PlGF) and vascular endothelial growth factor receptor-1 (VEGFR-1) in the study groups.

Localization	Staining Intensity	CHD	Control	*p*-Value *
a. Placental growth factor (PlGF)				
Syncytiotrophoblast	Absent staining, *n*	2/20	0/20	0.0046
	Weak staining, *n*	15/20	8/20	
	Moderate staining, *n*	2/20	8/20	
	Strong staining, *n*	1/20	4/20	
Decidual cells	Absent staining, *n*	4/20	0/20	0.0067
	Weak staining, *n*	15/20	14/20	
	Moderate staining, *n*	1/20	3/20	
	Strong staining, *n*	0/20	3/20	
Villous endothelial cells	Absent staining, *n*	10/20	8/20	0.8284
	Weak staining, *n*	8/20	10/20	
	Moderate staining, *n*	1/20	2/20	
	Strong staining, *n*	1/20	0/20	
b. Vascular endothelial growth factor receptor-1 (VEGFR-1)				
Syncytiotrophoblast	Absent staining, *n*	0/20	1/20	0.7069
	Weak staining, *n*	8/20	5/20	
	Moderate staining, *n*	5/20	10/20	
	Strong staining, *n*	7/20	4/20	
Decidual cells	Absent staining, *n*	1/20	0/20	0.2620
	Weak staining, *n*	7/20	3/20	
	Moderate staining, *n*	5/20	10/20	
	Strong staining, *n*	7/20	7/20	
Villus endothelial cells	Absent staining, *n*	3/20	3/20	>0.9999
	Weak staining, *n*	11/20	12/20	
	Moderate staining, *n*	6/20	4/20	
	Strong staining, *n*	0/20	1/20	

CHD, congenital heart defects. * Cochran–Armitage test for trend.

## Data Availability

The data that support the findings presented in this study are available from the corresponding author upon reasonable request.

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
