# Peer review of "Placental Pathology and Placental Growth Factor (PlGF)/Vascular Endothelial Growth Factor Receptor-1 (VEGFR-1) Pathway Expression Evaluation in Fetal Congenital Heart Defects"

_life, 2025, doi:10.3390/life15060837_

Round 1

Reviewer 1 Report

Comments and Suggestions for Authors

This is overall a well-written clinical research article. The authors begin by briefly delineating the complexity of likely pathogenic mechanisms involved in congenital heart defects (CHD), as normal heart development depends on many genes, some of which are also transcribed in the forming placenta in the same early stages of the first gestational trimester.

They evoke a “heart-placental” axis (but not whether there may be other, particularly cephalic, structures involved in this “axis”). The term is repeated at the introduction to the Discussion section, but a review of this concept if it is a broadly used one, could be useful for the reader as a reference. This evocation justifies the authors’ investigation into whether or not there is a correlation between CHD and placental anatomical or pathological defects, in particular concerning the expression patterns of PlGF and VEGFR1. This is an original and interesting perspective to pursue.

For this study, they choose to compare placentas from 51 pregnancies with fetuses bearing CHDs, to placentas from 51 stage-matched pregnancies without CHDs. The statistical investigations and comparisons of the case and control groups in part 3.1 are well conducted.

  1. The authors provided supplementary materials, which were welcome. However, I believe they should also add a file in the form of an Excel file that contains all the patient data, so that other researchers could reuse or reanalyze their data in other contexts. One might expect to see cleft lip/palate associated with the Tetralogy of Fallot cases, for example. This should contain anonymized information but cover the parameters mentioned in 3.1 and associated Table, to wit:

Maternal age, nulliparosity, any specific additional congenital defects where relevant (including kidney or other in addition to the CNS defects), fetal sex (not “gender”), fetal weight, fetal CRL, fetal HC and specific CHD diagnosed where relevant.

  1. I appreciated that the authors included scale bars for their Figure 2. However, they indicate in this figure that these are “representative micrographs”. The stainings appear convincing, but it would be even more so to include in the supplementary materials, images of these next to their adjacent control sections without the primary antibody but simply incubated with a control serum of the same species at the same time of processing, as they state they have done in the Materials and Methods section. It would of course be necessary to show in this supplement, that the images were treated in the same manner if corrected. And the authors could also indicate in the legend, those images classified as “weak”, “moderate” or “strong” for a visual reference.
  2. What is the importance of opening with this observation on line 249, “PlGF displayed a significantly weaker overall intensity of the immunostaining in all the investigated structures, compared to VEGFR-1 (Table 4).” As well as Figure 3? Given that the epitope is a secreted factor and not a membrane-bound protein, aren’t tissue compartment staining differences to be expected? I would rather see this sort of visual representation comparing CHD cases to controls for the different compartments in reference to the paragraph beginning on line 264, related to Table 5.
  3. Given that the authors compare Figure 3a and 3b, they should establish the same enlargement of these photographs at equivalent resolution (in particular, 3b). It would help the reader to add column legends and row names with the tissues and antibody targets shown.
  4. The sentence beginning line 253 is not an accurate statement. The authors may not discuss “staining intensity” without making quantitative measurements that compare the localization in the different cell types relative to a baseline, similar type of membrane-bound receptor stain. They need to be more explicit on what basis they make this assertion. In particular, although “Sections were randomized and blindly analyzed by a specialist” is the right approach, the authors should validate their findings by having another specialist make similar assessments of the staining intensities, and describing the disparities if any. This would make the conclusions much more solid.
  5. The authors are encouraged to rewite lines 316-323 with simpler sentences and perhaps more concision. Furthermore, at line 326-327 it is written that the “fetal-to placental weight ratio was lower, while placental-to-fetal weight ratio was higher in the CHD group”. These are not two separate observations though the sentence construction implies they are. Pick a ratio and use just that one, please. Lines 328-339 do not bring any useful information about the roles of PlGF/VEGFR1 and can be removed to help clarify as well.
  6. Line 386, why “In addition”? Is this finding concerning calcifications in contrast to something in the preceding paragraph? What is the authors’ interpretation of it?
  7. Line 417-419, for the reader, coming across “Wu et al. discovered that the CpG island inside the promoter of PGF gene was hypomethylated in both fetal growth restriction placentas and controls, without obvious differences between the two groups [44], ….” is a very surprising sentence in this context, until the very end: “thus offering a possible explanation for the weaker staining pattern.” I disagree with this supposition and suggest it be removed altogether (hypomethylated promoters tend to be associated with higher transcription, but this may not be associated with more protein that is detectable by the antibody). It is too speculative.
  8. Similarly, just afterward, the authors write in a sentence that should be cut in two at least, lines 423-427. It implies that VEGFR1 positively regulates the availability of PlGF transcript? Is this cell-autonomous?
  9. The authors could reduce their Discussion section by a page or two and not lose any of the interest of their work. I suggest they cut it down.

Minor comments:

  1. Line 57-59: The sentence is not very clear; heart development is far from being finished at 21d after fertilization in the human embryo. The choice of this specific moment to refer to heart tube fusion on one hand and ongoing villous formation on the other reads as arbitrary.
  2. Line 67-70: The observation of shared transcripts itself is not an argument to support their functional contribution to similar processes – these could just as well correspond to housekeeping proteins. The rest of the second introductory paragraph, though, reads as more solid.
  3. Line 79: “It determines its effects” is unclear: what determines its effects?
  4. It could be helpful to simply rephrase lines 87-92. In the introduction it was not clear if the controls in the study were also interruptions of pregnancy at the same fetal stages. However, this was immediately addressed subsequently in the Methods section 2.1.
  5. Figure 1 currently reads in terms of percentages, but it would be more natural to read in terms of number of cases, with the percentages in parentheses as n (%), as was done elsewhere.
  6. The authors begin to present placental pathology by starting with distinctions in the umbilical cord, which for the uninitiated, is surprising. Suggestion to change the section title to: “Morphological characteristics of placental and umbilical tissues”
  7. In Tables 2-5, please add the statistical test used to determine significant difference to the header or legend, along with p-value.
  8. Line 231, “With the exception of calcifications, that” > “With the exception of calcifications, which”
  9. Line 270, misspelling of “atrioventricular sepal defect” which should in any case be abbreviated to AVSD by this point.
  10. Line 288 should be “mother’s” in this sentence as written.
  11. Line 292: “These results are confirmed by other researchers in different studies” > “These results confirm those reported by other researchers.”
  12. Line 296: “Since the median gestational age of the subjects included in our study was lower compared to the abovementioned studies, we can conclude that the interactions between the abnormally developed heart and the placenta contribute to altered somatic growth in earlier stages of fetal development.” > “Since the median gestational age of the subjects included in our study was lower than in the abovementioned studies, we can conclude that interactions between an abnormally developed heart and the placenta contribute to altered somatic growth by earlier stages of fetal development.”
  13. Line 315, “ratio, few studies are using” > “ratio; a few studies use…”
  14. Please rewrite the sentence at lines 354-356 to simplify it.

Author Response

Thank you very much for taking the time to review our manuscript. Please find the detailed responses below and the corresponding revisions/corrections in track changes in the re-submitted files.

Response comment 1: The data presented in this study are available upon request from the corresponding author due to them being obtained from the registries of the Laboratory of Pathology, IMOGEN Research Center, County Emergency Clinical Hospital Cluj-Napoca, with the permission to be used for research purposes, but not for the publishing of raw data. Therefore we cannot publish the data as a Supplementary material, but we included them as Non-publishable material associated with the revised version of the manuscript, for the purpose of the peer-review process.

Response comment 2: We included in the Supplementary Materials the corresponding negative control (without the primary antibody) images, as suggested (Figures S1 and S2), while also adding the staining classification of the displayed slides in the respective figure caption.

Response comment 3: We agree with your observation. Due to the fact that one epitope is a secreted factor and the other a membrane-bound protein, staining differences and also tissue compartment differences are to be expected. The staining intensity differences between VEGFR-1 and PlGF are reported in previous studies, as mentioned in the Discussions chapter, while tissue compartment differences were not previously clearly stated, so we considered useful to emphasize them by statistical analysis. There are some differences between the way we imagined presenting the data between Table 4 and Figure 3. Table 4 compared the patterns of staining intensity between PlGF and VEGFR-1 in each of the three specific tissue compartments studied, while Figure 3 compared those staining patterns between the three compartments for each particular epitope. Also, for the comparison between CHD cases and controls for the different compartments, we chose the data representation in the form of a table (Table 5), not a graph, because we considered that it brings more clarity for the reader.

Response comment 4: We corrected the resolutions of the two parts of Figure 3 in order to be equivalent and we hope they meet the required criteria. Regarding the relationship between them, we did not intend to compare Figure 3a with Figure3b, each part of the figure was trying to represent a comparison between the three tissue compartments in terms of pattern of staining intensity for one of the specific epitopes studied. Labels were added to specify the tissue compartment represented in each column, and a legend with color codes for the specific staining intensities was also added next to the graph. The specific epitope is mentioned both on the graph (as a label of Y axis), and in the figure’s caption.

Response comment 5: We agree that IHC evaluation of expression is not the most sensitive and specific and does not offer accurate quantitative results. Due to technical and financial limitations, it was the assay of choice for this pilot study. The intensity of the immunostaining was evaluated by a 4-level semiquantitative scale, form “no staining” to “strong staining”, in the absence of more accurate means of evaluating staining intensity, such as imaging analysis software. We considered your suggestion and we validated our initial assessment independently by another pathology specialist, also randomized and blinded, using the same 4-level scale. The concordance between the two assessments was 84.2%, with the differences settled by the common evaluation of the slides by both specialists, reaching an agreement regarding the classification. We corrected Tables 4-5 and S1-S5 according to the combined data, but the overall results remained the same after the validation by the second specialist.

Response comment 6: We rewrote the whole paragraph to enhance its clarity.

Response comment 7: No, the finding concerning calcifications is not in contrast to something in the preceding paragraph, it is another observation derived from our dataset, hence the use of the expression “In addition”, that was intended to link this finding to the general findings regarding placental microscopic lesions in the context of fetal CHDs. Placental calcifications were associated with high-risk pregnancies, particularly with preeclampsia and fetal growth restriction, but previous studies display conflicting results. We considered that this observation was worth mentioning in the context of the unknown physiological ramifications of placental calcifications.

Response comment 8: We agree with your observation, so we removed this sentence.

Response comment 9: According to the work of Wu et al. (ref. 41 in the revised manuscript), yes, it is suggested a possible positive feedback regulation of PlGF expression.

Response comment 10: We adjusted the Discussion sections in order to reduce its length and retain the reader’s interest.

Response minor comment 1: We agree that heart development is far from being completed by the 21st day after fertilization and it is not what we intended to suggest in the abovementioned paragraph. We stated that the formation heart tube, as a functional primordium of the organ, is completed by that time, not the development of the whole heart. Courtney et al (ref. 7 from the revised manuscript) compared the parallel development of the heart and the placenta. In the cited article, the authors chose day 21 as a time-frame reference because at this point, the fusion of the heart tube is completed and also usually starts beating around the same time, with looping of the heart tube commencing in day 22-23; in the same time, around day 21, in the placenta, the tertiary villi are also developed enough. The authors wanted to emphasize the fact that around this time point the placental villous system is able to supply the beating heart tube with oxygen and nutrients, and that the relationship between the two organs is suggested even from the earlier embryonic development.

Response minor comment 2: We agree that the common transcripts do not imply common contribution to similar processes, but it still constitutes a valid observation that some of the commonly transcribed genes could have contributions to both organs’ development.  

Response minor comment 3: We rewrote the sentence to enhance the clarity.

Response minor comment 4: We edited the paragraph. We did not want to include methodological aspects in the paragraph stating the aims and objectives of the study, hence the choice of including more clear criteria for the control group in the Methods, section 2.1.

Response minor comment 5: We edited Figure 1 and added also the number of cases for each group, as suggested.

Response minor comment 6: We agree and changed the subtitles according to the suggestion.

Response minor comment 7: We explained in section 2.5. the teste used for the statistical analysis. For quantitative data, we used Mann-Whitney test, and for qualitative ones, Fisher’s exact test. For Tables 1-3, we added an explanation regarding the use of paired Wilcoxon test, where appropriate. Also, we added the use of Cochran–Armitage test for trend in the explanation for Tables 4-5 and S1-S5 in the Supplementary material.

Response minor comments 8-14: We corrected and rephrased the specific issues highlighted.

Reviewer 2 Report

Comments and Suggestions for Authors

In this study, the authors observed gross placental abnormalities and a lower placental IHC expression of PlGF in pregnancies with CHD fetuses. These findings contribute meaningfully to our understanding of the pathogenesis of CHD. I have two concerns:

1, IHC is not sensitive and specificity enough. Western blot analysis would provide more quantitative and specific protein-level data.

2, The mRNA expression levels of the two genes need to be detected to determine whether the observed protein-level changes reflect transcriptional regulation.  

Author Response

Thank you very much for taking the time to review our manuscript. We agree that IHC evaluation of expression is not the most sensitive and specific method and does not offer the most accurate quantitative results. Due to technical and financial limitations imposed, we considered the design of this pilot study using IHC as an assessment method, in order to have some preliminary data regarding this area. We are planning to validate our preliminary findings with more accurate expression analysis methods, including Western Blot and mRNA expression levels, in future studies, depending on funding approval. 

Reviewer 3 Report

Comments and Suggestions for Authors

Journal: Life (ISSN 2075-1729)

Manuscript ID: life-3620360

Type: Article

Title: Placental pathology and placental growth factor (PlGF) / vascular endothelial growth factor receptor-1 (VEGFR-1) pathway expression evaluation in fetal congenital heart defects

 Dear authors,

I had a pleasure reviewing your manuscript concerning placental pathology and placental growth factor (PIGF) / vascular endothelial growth factor receptor-1 (VEGFR-1) pathway expression in fetal congenital heart defects (CHD).

Fetal anomalies are always in the focus due to the significance of timely diagnosis, but, also, due the fact that etiopathogenesis is usually unknown.

Your work is, therefore, of a great significance.

The manuscript is excellent and I consider it worth publishing.

Author Response

Thank you very much for taking the time to review our manuscript and for your positive evaluation. We appreciate your kind words.

Round 2

Reviewer 2 Report

Comments and Suggestions for Authors

The authors need to notify the limitation of IHC in Discussion section.

Author Response

Thank you for your response. We updated the Discussion section to include a more detailed description of the limitations of IHC.